# Student Perception of Knowledge and Skills in Pharmacology and Pharmacotherapy in a Bachelor's Medical Curriculum

**Rahul Pandit \*** [ID]**, Merel C. S. Poleij and Mirjam A. F. M. Gerrits**

Department of Translational Neuroscience, Brain Center Rudolf Magnus, University Medical Center Utrecht (BCRM-UMCU), Utrecht University, 3584 CG Utrecht, The Netherlands
\* Correspondence: r.pandit@umcutrecht.nl

**Abstract:** Background: Pharmacology and pharmacotherapy (P&PT) is a foundational subject within the medical curriculum, preparing students for safe prescribing. The characteristics of students entering medical school change with time, and novel insights on teaching and learning also become available. A periodic review of the curriculum is required to investigate whether the current P&PT teaching optimally supports learning. Methods: To investigate this, the students' perceptions of their knowledge and competence in various P&PT topics were studied. A total of 152 third-year bachelor's students were invited to answer a 40-point online questionnaire. Results: The response rate for completing the questionnaire was 32% (N = 49). Students valued P&PT teaching, did not skip P&PT topics and desired more P&PT classes. Interestingly, students were hesitant to use recommended literature and textbooks to prepare themselves for classes. Concerning perceptions of knowledge and competence, students rated lower confidence in prescription writing skills and knowledge of drugs acting on the central nervous system. Conclusions: Although there are many positive elements within the current curriculum, the incorporation of teaching methodologies to ensure active student engagement is warranted. These modifications are essential to properly training the current generation of medical students for their role as future prescribers. A relatively low response rate and overestimation of one's competencies remain potential biases in the study.

**Keywords:** pharmacology; pharmacotherapy; teaching pharmacology; student perception

## 1. Introduction

Pharmacology and pharmacotherapy (P&PT) is one of the foundational subjects within the bachelor's medical curriculum. Knowledge about various drugs (pharmacology) and their application to treat patients (pharmacotherapy) is indispensable for medical doctors [1]. P&PT education begins early in the bachelor's phase in order to sufficiently prepare medical students for their role as prescribers. An early start together with regular repetition is especially important [2], as errors during prescribing are common [3] and are known to contribute to unnecessary hospital admissions [4].

During the bachelor's phase, students learn about individual drugs, their mechanism of action (effect), their side effects and possible interactions with other drugs [1]. Students additionally learn how drugs are processed in the body and how diseases could affect these processes [5]. Even though bachelor's students are not directly involved in real-world prescription writing, P&PT education in the bachelor's curriculum prepares medical students with the necessary knowledge and skills to do so during clinical clerkships and beyond. Interestingly, inadequacies within the undergraduate curriculum are known to contribute to prescribing errors [6]. Therefore, evaluation and optimization of P&PT teaching at regular intervals are essential to ensuring safe prescribing practices for future medical graduates.

Within the bachelor's program, students are taught P&PT-related topics in an integrated manner (the P&PT track), meaning P&PT-related topics are taught and examined

within separate thematic units alongside other preclinical and clinical disciplines [2]. This is in contrast to the traditional medical curriculum, where preclinical subjects such as pharmacology, anatomy, or physiology are taught as individual disciplines with independent final exams. The shift from a traditional to an integrated medical curriculum has been made by several medical faculties, as it aids students in understanding the interrelationship between various disciplines using clinical cases [7]. This is especially important for functioning in a clinical context, when knowledge of various disciplines has to be combined [8]. Despite these benefits, chances exist that students perform poorly in P&PT topics or minimally participate in P&PT education but still graduate, which is a known concern of educators [9]. Evaluating student knowledge, specifically in P&PT, is therefore essential and should be an ongoing process [10]. However, besides testing factual knowledge, asking about students' perceptions of their own competencies is equally important and can drive educational innovation. This approach has been studied for various medical disciplines, including pharmacology [11–13]. For example, final-year medical students are known to feel unprepared when it comes to prescribing drugs [14], and an adjustment of P&PT teaching has been suggested to address this issue [5]. Unfortunately, most of these studies are restricted to the master's phase, and little is known about how bachelor's students perceive their competence in P&PT-related topics. Inquiring students about their perceived level of knowledge and skills in P&PT will be helpful in guiding their own learning process [15]. This information is crucial for the timely correction of knowledge and skill-related deficiencies.

Another reason for evaluating the curriculum at regular intervals is to address the changing needs of the student population [16,17]. For example, in contrast to previous generations, the current generation of medical students needs more support while learning and prefers plenary lectures to group exercises. Therefore, understanding whether the current methods of instruction seamlessly align with the learning habits and preferences for the learning activities of the new generation of students is important [5]. Although feedback from standard course evaluations could trigger curricular changes, such feedback is unfortunately superficial, nonspecific and mostly measures student satisfaction [18,19]. These evaluations are deemed unsuitable by teachers for eliciting educational changes [19]. As a result, their use for this purpose is limited [20]. Deeper reflection into one's own motivation to study various subjects or preferences for learning activities is not assessed in most evaluations. Additionally, due to the fragmented nature of P&PT teaching within the integrated medical curriculum, only specific topics within P&PT are evaluated at a given time instead of the entire track. This makes it increasingly difficult to gain insight into P&PT education at the University Medical Center Utrecht (UMCU). A detailed evaluation of the entire track is therefore vital to highlighting the strengths and weaknesses of the P&PT education in the bachelor's phase.

In view of this, the current study aims to thoroughly evaluate the P&PT track within the bachelor's medical curriculum. Based on these results, P&PT education can be improved to optimally support student learning and thereby contribute towards safe prescribing.

The methodology of the current study could be used to shift from evaluations merely measuring student satisfaction to ones that are specific and incorporate student reflection on desired topics. The findings of the current study could not only be relatable for P&PT teachers; the recommended alterations in P&PT teaching could additionally guide curricular changes to ensure safe prescribing practices.

## 2. Materials and Methods

### 2.1. Ethical Approval

Ethical approval for the research was granted by the Dutch society for medical education (NERB 2020.1.3).

### 2.2. Study Setting

The current study took place at the UMCU in the Netherlands. P&PT education is embedded within individual courses across all years of the 3-year bachelor's medical

program. The individual courses run for a duration of 3–8 weeks and are followed by a course assessment.

### 2.2.1. Teaching Modalities

The P&PT curriculum is built up using the following teaching modalities:

- Plenary lectures (300 students): For this teaching form, no prior preparation is usually required as the teacher explains the P&PT topics. Students may ask questions to clarify doubts during the course of the lecture or at the end. Plenary lectures are recorded and followed up by interactive lectures, seminars or small group teaching.
- Interactive lectures (75–150 students): During interactive lectures, answers to self-studies are discussed in an interactive fashion. Active participation is stimulated. Students are required to read excerpts from books or solve clinical cases individually or in groups before attending the classes. These sessions are not recorded.
- Seminars (36–60 students) and small groups (12–24 students): Depending on the logistics (availability of time slots and teacher availability), group sizes are determined for an in-depth discussion of case studies. Prior preparation is required for these classes. None of the P&PT lessons are compulsory for students, but an overall attendance of 80% for small group teaching is advised.

### 2.2.2. Preparation for Attending P&PT Lessons

For all lessons, with the exception of plenary lectures, students are required to either study chapters from books on pharmacology, watch relevant videos or follow e-modules to complete self-study assignments. These assignments are either provided in the course manual or are shared on the online platform Blackboard (www.blackboard.com). The platform PScribe (www.pscribe.nl) was used for teaching prescribing skills. Whether students prepare for the lessons is not formally checked.

### 2.2.3. Assessments

Assessment of P&PT topics occurs as a part of the final course exam. Questions on P&PT topics are embedded within the final course exam, comprising closed-ended or open-ended questions [10]. Students do not need to pass P&PT as a discipline in order to successfully complete the course.

Table A1 in Appendix A provides an overview of the obligatory bachelor's courses where P&PT is taught and includes the topics, teaching modalities and hours devoted to these topics.

### 2.3. Study Setup

Questionnaire: The study was conducted as a survey using a 40-point questionnaire that consisted of four domains aimed at gaining insight into the various aspects of P&PT education. The questionnaire was tailored to evaluate the P&PT curriculum at the UMCU but was inspired by previous research on the same topic [11–13,21]. The questions were divided into four categories, designed to inquire about the three elements from the theory of constructive alignment [22] and student motivation to study P&PT [23].

1. The level of student motivation to study P&PT: This category of seven questions inquired on students' intrinsic motivation to study P&PT and was based on the self-determination theory [23]. This set of questions was incorporated as it is known that many students find P&PT challenging [24] and might therefore be unwilling to study the subject, especially since it only partially contributes to the course.
2. The extent of self-perceived competence in various P&PT topics: This category was structured into three subcategories and inquired about students' perceptions of their own knowledge or skills relating to specific drug groups (subcategory A, nine questions), how drugs act and are processed in the body (subcategory B, three questions) and specific steps in drug prescribing (subcategory C, eight questions). This category

of questions is devised to provide answers to specific P&PT themes relevant for future prescribing practices.

3. The reflection on the employed methods of examination: The third category consisted of three questions focusing on the methods of examination employed to evaluate learning in P&PT.

4. The reflection on employed P&PT teaching methods: The final category consisted of 10 questions and aimed to acquire information on student perceptions of teaching methods currently being used and their openness to other novel teaching methods in the future.

Students had to indicate their level of agreement or disagreement with statements based on a 7-point Likert scale (1 = strongly disagree, 2 = disagree, 3 = slightly disagree, 4 = neutral, 5 = slightly agree, 6 = agree, 7 = strongly agree). As suggested in the literature [25], prior to the dissemination of the questionnaire, various groups were consulted to provide feedback on the formulation and interpretation of the statements. For this, two students with a similar profile as the study population, a nonteaching supporting staff member, an expert on survey-based research and two educational experts were approached.

Student population: A random group of third-year medical students enrolled in a preparatory course prior to starting their first clerkship was invited to fill out the questionnaire between October 2020 and October 2021. The preparatory course is offered several times a year, and students from a single academic year were approached to fill out the online questionnaire. This group was specifically selected as students following this preparatory course are required to have successfully completed the majority of first- and second-year courses, including courses with P&PT topics, which would enable them to reflect on the entire P&PT curriculum.

Survey dissemination: Students were invited by email by a nonteaching supporting staff member to fill out the questionnaire during the ongoing preparatory course. The nature of the survey and the pros (e.g., contributing towards improving P&PT education) and cons (e.g., time investment) of participating in the study were shared. Students had to give their informed consent prior to participation. A single reminder was sent to students one week following the initial survey dissemination, following which the survey was closed after three weeks. Students received no incentives for filling out the questionnaire.

Data collection and analysis: The data were anonymously collected using the program Qualtrics. No personal information such as age, gender or ethnic background was recorded. All data, including that from incompletely filled questionnaires, were used for the analysis. The data were extracted from Qualtrics and analyzed using Microsoft Excel and SPSS. The Cronbach alpha was computed for the entire questionnaire.

## 3. Results

A total of 152 students were invited to participate in the survey, of whom 63 (41%) filled out the questionnaire partially and 49 (32%) completed it entirely. The Cronbach's alpha value for the entire questionnaire showed the questionnaire to reach acceptable reliability, $\alpha = 0.85$. Removal of individual items resulted in minor changes in Cronbach's alpha value (0.84–0.86), and all items were thus retained.

The findings of the survey are summarized in Table 1. The majority of the students were motivated to study and valued P&PT education, and students recognized the relevance of P&PT in the context of their future work settings. In general, students prepared themselves for P&PT lessons. Strikingly, a substantial number of students did not use the text books recommended for this purpose. In relation to self-perceived knowledge of the various drug groups, students were least confident in drugs acting on the central nervous system (antidepressants and benzodiazepines), followed by drugs used for ischemic heart diseases. Although self-perceived knowledge of pharmacokinetics (how drugs are processed in the body) and pharmacodynamics (how drugs act on the body) was high, students felt less competent when it came to certain aspects of rational pharmacotherapy. For example, writing prescriptions and choosing drug dosages scored low on the Likert scale.

**Table 1.** Student perceptions about various aspects of P&PT education. The number of students answering a group of statements is indicated in brackets (N). Negative statements are indicated with a †. Frequencies for each Likert scale option, mean (ME) and median (MD), are provided for each statement. Likert scale key: SD: strongly disagree; D: disagree; SLD: slightly disagree; N: neutral; SLA: slightly agree; A: agree; SA: strongly agree.

| No. | Statements | Likert Scale | | | | | | | Statistics | |
|---|---|---|---|---|---|---|---|---|---|---|
| | | SD (1) | D (2) | SLD (3) | N (4) | SLA (5) | A (6) | SA (7) | ME | MD |
| | The level of student motivation to study P&PT (N = 63) | | | | | | | | | |
| 1 | I enjoy P&PT lessons | 0 | 0 | 0 | 3 | 13 | 38 | 9 | 5.8 | 6 |
| 2 | I consider P&PT an interesting subject | 0 | 0 | 1 | 4 | 12 | 37 | 9 | 5.8 | 6 |
| 3 | I do my very best for P&PT | 0 | 0 | 2 | 5 | 8 | 42 | 6 | 5.7 | 6 |
| 4 | During a course, I always leave out P&PT topics † | 29 | 21 | 8 | 0 | 4 | 1 | 0 | 1.9 | 2 |
| 5 | I always prepare myself for P&PT classes | 0 | 3 | 11 | 5 | 17 | 23 | 4 | 4.9 | 5 |
| 6 | I think that P&PT is important for my future profession as a doctor | 1 | 1 | 0 | 1 | 2 | 25 | 33 | 6.3 | 7 |
| 7 | I always read the recommended books and literature to prepare for P&PT classes | 10 | 18 | 15 | 12 | 6 | 2 | 0 | 2.9 | 3 |
| | Extent of self-perceived competence in various P&PT topics | | | | | | | | | |
| | Specific drug groups (N = 53): I think I am competent * in the following P&PT topics. * I can explain the mechanism of action, the most important side effects and contraindications of these drug groups without using any formularies or books. | | | | | | | | | |
| 8 | Antidepressants | 4 | 10 | 15 | 8 | 11 | 5 | 0 | 3.5 | 3 |
| 9 | Antibiotics | 2 | 9 | 4 | 5 | 16 | 13 | 4 | 4.5 | 5 |
| 10 | Antihypertensive drugs | 2 | 2 | 0 | 3 | 18 | 24 | 4 | 5.3 | 6 |
| 11 | Drugs for ischemic heart diseases | 2 | 7 | 10 | 12 | 10 | 11 | 1 | 4.1 | 4 |
| 12 | Drugs affecting blood coagulation | 1 | 3 | 5 | 6 | 22 | 14 | 2 | 4.8 | 5 |
| 13 | Benzodiazepines | 3 | 10 | 19 | 7 | 12 | 1 | 1 | 3.4 | 3 |
| 14 | Drugs for treating pain | 2 | 3 | 3 | 3 | 23 | 16 | 3 | 4.9 | 5 |
| 15 | Antidiabetic drugs | 1 | 2 | 6 | 4 | 22 | 15 | 3 | 4.9 | 5 |
| 16 | Bronchodilators | 1 | 3 | 1 | 4 | 18 | 18 | 8 | 5.3 | 5 |
| | How drugs act and are processed in the body (N = 51): I think I am competent in the following P&PT topics: | | | | | | | | | |
| 17 | Understanding of pharmacokinetic concepts (absorption, metabolism, distribution and elimination of drugs) | 0 | 0 | 0 | 2 | 12 | 27 | 10 | 5.9 | 6 |
| 18 | Understanding of pharmacodynamic concepts (drug-receptor binding and mechanism of action of drugs) | 0 | 1 | 1 | 3 | 18 | 22 | 6 | 5.5 | 6 |
| 19 | Understanding the clinical consequences of up or down regulation of receptors | 0 | 0 | 0 | 1 | 8 | 26 | 16 | 6.1 | 6 |
| | Specific steps in drug prescribing (N = 49): You are following a clinical clerkship under the guidance of the general practitioner. During a consultation, you see a patient with a particular disease. You have sufficient knowledge of this disease. For example, stable angina pectoris, type 2 diabetes mellitus, atrial fibrillation, hypertension, pneumonia, COPD<, osteoarthritis or bronchial asthma. Your supervisor asks you to formulate a treatment plan following the 6-step WHO method. You are allowed to consult formularies. | | | | | | | | | |
| | I think I am competent in the following P&PT topics: | | | | | | | | | |
| 20 | Independently choose the correct drug for the patient | 0 | 1 | 2 | 1 | 13 | 26 | 6 | 5.6 | 6 |
| 21 | Independently choose the appropriate dosage form | 0 | 2 | 1 | 3 | 20 | 20 | 3 | 5.3 | 5 |
| 22 | Independently choose the appropriate dose | 3 | 9 | 11 | 4 | 13 | 8 | 1 | 3.9 | 4 |
| 23 | Modify treatment choice based on drug-drug interactions | 0 | 7 | 4 | 9 | 15 | 10 | 4 | 4.6 | 5 |
| 24 | Modify treatment based on the hepatic and renal functions | 0 | 2 | 2 | 7 | 12 | 19 | 7 | 5.3 | 6 |
| 25 | Independently write a prescription | 6 | 11 | 12 | 6 | 10 | 4 | 0 | 3.3 | 3 |
| 26 | Independently inform the patient about the details of drug administration (expected effects, instructions for drug use and possible side effects) in simple language | 0 | 1 | 4 | 5 | 15 | 19 | 5 | 5.3 | 5 |

**Table 1.** *Cont.*

| No. | Statements | Likert Scale | | | | | | | Statistics | |
|---|---|---|---|---|---|---|---|---|---|---|
| | | SD (1) | D (2) | SLD (3) | N (4) | SLA (5) | A (6) | SA (7) | ME | MD |
| 27 | Independently formulate a plan to evaluate the efficacy of the treatment | 0 | 4 | 8 | 8 | 15 | 12 | 2 | 4.6 | 5 |
| | Reflection on the employed methods of examination (N = 49) | | | | | | | | | |
| 28 | P&PT education in the preclinical years has prepared me sufficiently for my clinical clerkships | 0 | 0 | 6 | 9 | 15 | 18 | 1 | 5.0 | 5 |
| 29 | While preparing for examinations, I always skip P&PT topics | 22 | 17 | 6 | 2 | 2 | 0 | 0 | 1.9 | 2 |
| 30 | I think that the current method of testing (digital and written) optimally tests my P&PT knowledge | 1 | 1 | 9 | 10 | 11 | 14 | 3 | 4.7 | 5 |
| | Reflection on employed P&PT teaching methods (N = 49) | | | | | | | | | |
| 31 | I learn a lot during small group teaching when P&PT knowledge is applied to solve case studies | 0 | 1 | 1 | 1 | 5 | 24 | 17 | 6.1 | 6 |
| 32 | I learn a lot during plenary lectures where pharmacological concepts are explained | 0 | 0 | 0 | 3 | 10 | 16 | 20 | 6.1 | 6 |
| 33 | I learn a lot during interactive lectures where self-study assignments are discussed | 0 | 4 | 2 | 4 | 8 | 18 | 13 | 5.5 | 6 |
| 34 | The internet program (PScribe) is a handy way to practice prescribing of drugs | 7 | 10 | 14 | 8 | 6 | 3 | 1 | 3.2 | 3 |
| 35 | I think that role-play in a simulated environment with students or actors would aid pharmacotherapy learning | 1 | 6 | 5 | 13 | 11 | 9 | 4 | 4.4 | 4 |
| 36 | I think that reading scientific literature (journal clubs) on pharmacological topics would increase my knowledge of P&PT | 1 | 8 | 7 | 12 | 16 | 5 | 0 | 4.0 | 4 |
| 37 | I am prepared to dive deeper into pharmacological topics by conducting a literature study | 1 | 7 | 7 | 7 | 15 | 11 | 1 | 4.3 | 5 |
| 38 | I wish we had more P&PT lessons | 0 | 0 | 3 | 10 | 15 | 17 | 4 | 5.2 | 5 |
| 39 | I would like more formative exams in order to be aware of my level of knowledge | 1 | 5 | 4 | 5 | 14 | 13 | 7 | 4.9 | 5 |
| 40 | I would like feedback from my peers in order to learn from each other | 3 | 7 | 8 | 17 | 5 | 8 | 1 | 3.9 | 4 |

Furthermore, students felt largely confident that P&PT education had appropriately prepared them for clinical clerkships and agreed that the current method of examination appropriately tested their knowledge. The majority of students did not skip P&PT-related topics while preparing for exams for integrated courses.

Finally, appreciation levels for all types of employed teaching settings (seminars, plenary sessions and interactive lectures) were reasonably high and equal. However, students seemed less enthusiastic when it came to using the electronic prescribing simulator (Pscribe). Regarding future changes in the P&PT curriculum, students indicated openness for literature studies and frequent formative examinations but were neutral towards P&PT sessions involving role-play, peer feedback and journal clubs.

## 4. Discussion

The ultimate skill of prescribing drugs is a product of the progressive acquisition of knowledge about the mechanism of action of drugs and understanding how drugs interact with the organism. To master this skill, a sound understanding of parallel disciplines such as cell biology, biochemistry, physiology, and clinical medicine is essential [24]. This makes P&PT an intimidating subject within the medical curriculum [24]. It also makes students vulnerable to knowledge deficiencies in this subject [26–28]. The current study provides valuable information on students' perceptions of their own competence on various P&PT topics and sheds light on how students experience P&PT teaching at the UMCU. Our study is distinct as it focuses on bachelor's students and shows that feedback from students,

incorporating elements of reflection rather than only measuring student satisfaction [19,20], could provide valuable information [26,29,30] to trigger curricular changes.

The high intrinsic motivation to study P&PT demonstrates students' willingness to learn and take responsibility for their role as future medical professionals. Openness to conducting literature studies on P&PT-related topics or taking formative exams further supports this notion. Although most students prepare for P&PT classes and do not leave out P&PT topics embedded within integrated courses, a substantial portion of the respondents skip reading necessary literature or books. This phenomenon could in fact reflect a generic characteristic of the current generation of medical students, where videos and shorter texts are preferred to longer reads or traditional books [16]. A shorter attention span among students potentially underlies this phenomenon [16]. Moreover, the current generation of medical students, the so-called digital natives, are known to search for online materials rather than use prescribed textbooks [17,27]. Although the future P&PT curriculum should offer teaching tools such as e-modules, shorter videos or mobile apps to deliver information, reading textbooks should still be encouraged for literacy development in medical undergraduates [16]. Especially since the current generation is adept at finding and assimilating information online, understanding the differences between facts and opinions becomes crucial [27]. A better integration of reading materials with shorter texts could be helpful in achieving this goal [16]. Based on these insights, the hours devoted to plenary lectures have been diminished in the ongoing curricular revision. Instead, short e-modules, lectures and video clips are being provided as preparatory materials. The subsequent P&PT lessons consist of briefly revisiting the P&PT principles, mostly discussing their application. For this, new clinical cases are handed out on the spot.

The feeling of unpreparedness regarding prescribing drugs, observed also in final-year medical students [14,28], is reflected in the current group of bachelor's students. The desire to receive more P&PT lessons could be a result of the underlying feeling of unpreparedness before starting clinical clerkships [31]. Alternatively, it could reflect the intrinsic motivation to study P&PT. For the P&PT curriculum, this means discussing with the medical faculty the potential expansion of teaching hours or teaching support for conducting small group teaching.

The current cohort of students especially struggles with writing prescriptions and choosing drug dosages, a finding that can be explained by the limited training offered in this area presently. Early initiation and continued training of prescribing skills are therefore essential to developing prescribing competence in later years [5,14]. Unfortunately, the use of the digital platform PScribe, which is ideal for training prescribing skills [32], is not overwhelming. For this, a follow-up evaluation inquiring into the reasons behind students' hesitation to use this platform is required. To strengthen prescribing skills, exercises using the platform PScribe, with a focus on prescription writing are being provided frequently. Despite this, we believe that prescribing medication is a skill that, like other medical skills, needs to be trained rigorously. Increasing the number of hours dedicated to P&PT and creating space for small group teaching, especially by involving simulation patients [33], would provide opportunities for practice and individualized feedback. This, however, requires drastic curricular change and would have financial consequences.

Finally, concerning the students' perception of their own knowledge of various P&PT topics, the subjective feeling of confidence in drugs acting on the central nervous system (antidepressants and benzodiazepines) was lower compared to other drug groups. The reason behind this finding in our cohort cannot be exactly pinpointed. Lack of repetition is a likely explanation, as these topics are offered only once within the bachelor's curriculum. Another reason could be the intrinsic difficulty of the subject matter, which is also witnessed in other schools [34]. That said, student confidence in other drug groups, although better, was not extremely high. Loss of knowledge is a known phenomenon in medical school and should be addressed. A spaced repetition approach [35] or using microlearning tools [36] could be effective strategies to improve student knowledge on these topics. As the courses where P&PT teaching is provided are 'fixed' within the curriculum, offering

additional P&PT classes or testing P&PT knowledge separately is not an option, especially in an overloaded curriculum. Instead, a strategy that has been implemented is the use of microlearning, where students receive multiple-choice questions on various P&PT topics via email from the platform Redgrasp (www.redgrasp.com). Questions are provided with an elaborate answer aimed at serving as a quick revision of these topics. Topics that students consider difficult are more frequently quizzed than other topics. Besides this, the Dutch national pharmacotherapy assessment has been introduced at the UMCU in the master's phase [32]. This is a national exam, similar to the prescribing safety assessment in the UK [37]. All Dutch medical undergraduates are required to take this exam prior to graduation. The aim of the exam is to ensure revision of all drug classes. Next to this, multiple P&PT topics are revisited in the master's phase, providing students with additional opportunities to sharpen their knowledge and skills.

Questioning students on their preferences for educational methodologies applied revealed equal preferences for both case-based learning in small groups and plenary lectures. This finding, although observed by some researchers [38], deviates from other findings where lectures are preferred to small group teaching [27]. Plenary lectures are essentially a passive way of acquiring knowledge [39]. Moreover, in the current setting, answers to case studies and lecture notes of former students are widely available, which reinforces passivity and negatively affects attendance at both lectures and small group lessons alike. Instead, methods encouraging active participation, such as team-based learning [40] or flipping the classroom [41], could be suitable alternatives. Team-based learning not only nudges students to prepare themselves for the class by administering readiness assurance tests, but it also trains students to solve problems related to their future field of work [40]. Similar to other universities [41,42], TBL is currently being introduced at the UMCU in a phased manner. Moreover, as working in teams is a core element of the medical profession, early exposure to these teaching techniques could be beneficial in the long run. This is especially true since the current generation of students prefers working independently rather than in groups [16]. This inclination plausibly explains the lower scores observed in our cohort with respect to willingness to learn P&PT through journal clubs or role-play enactment. Nevertheless, lower openness for these teaching methods does not necessarily justify their exclusion from the curriculum; rather, further inquiry on student hesitation for these teaching methods should be pursued. The pending major curricular change in most Dutch medical schools based on national recommendations on health care education [43] could provide an opportunity to design courses where these elements are incorporated into teaching.

### 4.1. Limitations

For a correct interpretation of the results, it is important to discuss the possible limitations of the current study. The overall response rate of 32% might seem low. However, these responses are similar to those in other surveys on medical education [38,44]. The relatively low response rates for online evaluations are common across other faculties and could have multiple reasons, such as survey fatigue and a lack of incentives or reservations about privacy or anonymity [45,46]. Similar to other surveys, bias due to nonresponse cannot be entirely excluded. Interestingly, it has been established that the number of nonresponders correlates poorly with nonresponder bias [47]. This suggests that bias due to nonresponse could have a minimal impact on study results. Methods to increase the number of responders, such as using personal authority as a teacher [48], were not applied to the current study as they could influence the results of the study or push students to give socially accepted answers [47]. Removing anonymity by providing incentives [45] would have similar effects and thus were not applied.

The current study population consisted of third-year medical students enrolled in a pre-clerkship preparatory training course. This time point was chosen as students following this course would have followed all courses where P&PT was taught, a prerequisite to being

able to reflect on the entire P&PT curriculum. A limitation, however, was that students who did not follow the course were automatically excluded.

Yet another concern could be the question of whether student perception correlates with actual knowledge of the surveyed P&PT topics. It is known that students can overestimate their knowledge, leading to overconfidence [49]. Since the anonymity of students was essential for honest feedback, individual responses could not be correlated with grades on specific topics. For this reason, this notion can neither be proved nor rejected. That said, inquiring about student perception to drive educational innovation has been used previously [11,29,34] and is a source of inspiration for the current study.

### 4.2. Implications and Future Direction

Over the years, the medical curriculum has shifted from a traditional discipline-based curriculum to an integrated curriculum where various disciplines are integrated in thematic courses aiming to reduce redundancy and promote a deeper understanding of biomedical topics [7]. However, completely bypassing or eliminating basic biomedical principles from the curriculum could be disastrous. For P&PT, this could mean using web-based formularies as 'cookbooks' to treat patients instead of guiding choices through logical deductive processes [50]. It is therefore the task of P&PT educators to optimally support students in order to prepare them as future prescribers.

We believe that a revision of the curriculum is essential to keeping in line with the changing characteristics of students and novel insights on educational methods. Changing the content of longitudinal tracks or threads such as P&PT within an integrated curriculum is a formidable task as it requires adjustments in teaching hours or modalities in other disciplines as well. Minor changes, such as incorporating a novel teaching method or embedding a quiz, are simpler to implement and should be routinely undertaken. Changing the entire teaching modality, for example, switching from a lecture to small group teaching, requires thorough planning, discussion of resources (teachers, availability of support staff) and negotiation with other disciplines. Although time-consuming, an open discussion about it with the faculty and the involved students is recommended. For the UMCU, the proposed curriculum, drawn on the national recommendations on medical curriculum [43], is ideal to initiate these changes, and the current results provide a strong basis to initiate this transition.

Finally, it is important to note that research in education is fundamental for any educational innovation, including individual disciplines such as pharmacology. A research-informed approach provides a systematic and grounded approach to guiding curricular change. Not only should the applied modifications in a curriculum be evidence-based, but how these modifications fare locally should also be studied accordingly. For the current study, it means that next to using student perception as a tool to guide curricular changes, measuring student performance on P&PT topics could be insightful. In fact, in another study, we show that first-year students have suboptimal knowledge in pharmacokinetics, a subdomain of P&PT [10]. An opportunity to do the same for various drug groups could be the Dutch Pharmacotherapy Exam [32], where knowledge of various drug groups as well as rational pharmacotherapy is examined.

**Author Contributions:** R.P.: conceptualization, methodology, formal analysis, writing the original draft. M.C.S.P.: analysis, review and editing. M.A.F.M.G.: conceptualization and review. All authors have read and agreed to the published version of the manuscript.

**Funding:** This research received partial funding from the University of Utrecht to stimulate scholarship of teaching and learning (SoTL).

**Institutional Review Board Statement:** Ethical approval for the research was granted by the Dutch society for medical education (NERB 2020.1.3).

**Informed Consent Statement:** Informed consent was obtained from all subjects involved in the study.

**Data Availability Statement:** The datasets generated during and/or analyzed during the current study are not publicly available but could be available from the corresponding author on reasonable request, provided prior permission is granted by UMC Utrecht.

**Acknowledgments:** We thank Lindy A. Wijsman, an educational consultant, for providing valuable feedback on the methodology of the study.

**Conflicts of Interest:** The authors declare no conflict of interest.

## Appendix A

**Table A1.** List of obligatory courses where pharmacology and pharmacotherapy (P&PT) are taught within the bachelor's medical curriculum.

| | Obligatory Courses | | P&PT Topic | Mode of Teaching |
|---|---|---|---|---|
| | | | Year 1 | |
| 1 | Form & Function | - | Introduction to pharmacology and pharmacotherapy | PL (1), IL (2) |
| 2 | Healthy and diseased cells | - | Signaling cascades | IL (2) |
| 3 | Metabolism I | - <br> - <br> - <br> - | Pharmacokinetics: Absorption and metabolism of drugs <br> Drugs for treatment of hyperacidity <br> Drugs for treating obstipation <br> Drugs for treating iron-deficient anemia | IL (2), SG (2) |
| 4 | Sense organs, brain and movement I | - <br> - <br> - | Drugs acting on the autonomic nervous system <br> Drugs acting on the central nervous system <br> Introduction to pharmacodynamics | PL (4), IL (2), SG (2) |
| 5 | Circulation I | - | Pharmacokinetics: Elimination of drugs | IL (2), SG (1) |
| 6 | Infection and Immunity I | - <br> - | Introduction to antibiotics <br> Drugs used for urinary tract infections | IL (1) |
| 7 | Regulation and Integration | - | Quantitative pharmacodynamics and pharmacokinetics in health and disease | PL (4), IL (6), SG (6) |
| 8 | Circulation II | - <br> - | Drugs acting on blood coagulation <br> Drugs acting on the heart, blood vessels and the kidneys | PL (3), S (2) |
| | | | Year 2 | |
| 1 | Sense organs, brain and movement 2 | - | Drugs used for treating pain | IL (2) |
| 2 | Circulation 2 | - <br> - | Drugs for treating systemic hypertension <br> Drugs for treating chronic pulmonary disorders | IL (2) |
| 3 | Healthy and diseased cells II | - | Introduction to chemotherapy | Incorporated in plenary lectures of discipline oncology |
| 4 | Metabolism II | - | Drugs for treating diabetes mellitus | IL (1) |
| 5 | Sense organs, brain and movement III | - | Drugs for treating dermatological disorders | Incorporated in plenary lectures of discipline dermatology |
| 6 | Growth and development | - | Drugs during pregnancy and lactation | IL (2) |

**Table A1.** *Cont.*

|   | Obligatory Courses | | P&PT Topic | Mode of Teaching |
|---|---|---|---|---|
| | | | Year 3 | |
| 1 | Healthy and diseased cells II | - | Chemotherapy of individual tumors | Incorporated in plenary lectures of discipline oncology |
| 2 | Medical humanities | - | Drug development and ethical aspects of drug use | S (2) |
| 3 | Blok Green | - | Clinical pharmacokinetics and dynamics | SG (4) |

The numbers in brackets in the last column indicate hours dedicated to P&PT. The list does not include hours devoted to preparation. Also, P&PT topics, especially drug groups, are also discussed during lectures in other disciplines and are not depicted in this table. These hours are not mentioned in the table. PL (plenary lecture, 300 students), IL (interactive lecture, 100–150 students), S (seminars, 36–60 students), SG (small groups, 12–24 students)

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
