# Peer review of "Student Perception of Knowledge and Skills in Pharmacology and Pharmacotherapy in a Bachelor’s Medical Curriculum"

_ime, doi:10.3390/ime2030020_

Round 1
Reviewer 1 Report
This study explores the perceptions of medical students regarding pharmacology and pharmacotherapy (P&PT) education, emphasizing the need for an effective curriculum that aligns with students' motivations and promotes their future prescribing competence. Some of the comments for the authors considerations are listed below
Comments for authors:
Title:
The title accurately reflects the content of the study and is concise and informative.
Abstract:
- Please follow the journal formatting and add the abstract section headings.
- With the limits imposed on the abstract a short sentence on the Limitations with a brief discussion of potential limitations, such as the response rate and potential bias in student perceptions, would add credibility to the study and help contextualize the results.
Introduction:
1. Rationale for P&PT Track: Providing a brief rationale for why P&PT topics are integrated into thematic units alongside other preclinical and clinical disciplines may help readers understand the context of the study better.
2. It would be beneficial to include a paragraph summarizing the potential contributions of the study to the field of medical education. This can highlight how the findings may lead to improvements in P&PT education and enhance prescribing practices.
Methods:
1. Cronbach Alpha: Provide the calculated value of Cronbach's alpha for the entire questionnaire. This will help assess the internal consistency and reliability of the survey instrument.
Results:
1. Participant Information: The section lacks information about the demographic characteristics of the participants, such as age, gender, and academic performance. Including this information would help contextualize the findings and assess potential biases.
2. The section could benefit from a brief discussion of potential limitations of the study, such as the response rate, selection bias, and the generalizability of the findings. This would provide readers with a more comprehensive understanding of the study's scope and implications.
Discussion:
1. The "Discussion" section could be enriched by providing specific recommendations for enhancing the P&PT curriculum at UMCU based on the survey findings. These recommendations could be linked to the identified knowledge deficiencies and preferences of the students.
2. The section could discuss potential strategies to address knowledge deficiencies in specific drug groups, such as implementing additional training or revisiting certain topics during clinical clerkships.
3. While the "Limitations" section mentions that the 32% response rate is comparable to other medical education surveys, further discussion or contextualization of this response rate in the context of similar studies could strengthen the point.
4. To address the concern of students potentially overestimating their knowledge, the section could suggest future research or strategies to validate student perceptions against actual knowledge, perhaps through objective assessments or longitudinal studies.
5. The "Implications and Future Direction" section could be expanded to include more specific recommendations for curriculum revision. Discussing potential changes, such as integrating basic biomedical principles while promoting deeper understanding or adopting new educational methodologies, would add depth to the discussion.
Conclusions:
There is no conclusion section
Author Response
Please find the reply attched as a word document.

Reviewer 2 Report
This is a description of students’ perceptions of knowledge and skills of medical students concerning pharmacology and pharmacotherapy at a single institution. Although the results are of utmost local importance, I could not see the generalizability of the results.
A major problem is that the authors do not describe their current curriculum in general and P&PT sessions in particular. As such, they conclude form the survey results that changes should be done (e.g. Lines 23-24 and 273-276) but we as readers cannot infer as what really works and what not. For example for teaching methods, they argue that there should be more active methods such as TBL… but I could not find the basis for that. The same applies for the conclusions on assessment.
The survey questions are at times confusing and the categories do not always make sense. To name a few: why is Q28 under examination and why is Q39 under teaching methods. Also, I am not sure for which items was the Cronbach alpha calculated. The way it should be done is by construct and I don’t see any.
Lines 184-189: unfocused and speculative.
Author Response

(The authors gave the same response as above.)

Reviewer 3 Report
In this study, Pandit et al. performed a questionnaire to third-year bachelor medical program students to access their perception of knowledge and competence regarding pharmacology and pharmacotherapy (P&PT). Their findings suggest that the students were positive to their knowledge and competence in P&PT (except drugs affecting the central nervous system) but needed a more engaging and active learning. Overall, the findings are interesting, and the manuscript is well written. However, there are some concerns that should be addressed.
1. It is not clear if the students were all from the same class or different classes since the questionnaire collection was from October 2020 to October 2021. Or did the students had 1 year to fill the questionnaire. This should be clarified in the methods section.
2. It would be more informative to the study if the authors have any information why only 32% of the students answered the questionnaire. Could that lead to a bias in the analysis with only the motivated students in P&PT answered? This should be address in the limitation section.
3. The mean of answers for questions 24 and 25 might need to be revised.
4. The authors noticed that the students feel less confident with drugs affecting the central nervous system than the other topics. Can this relate to the teaching format that is different from the other topics? A lack in the physiology/anatomy of the central nervous system? It would be interesting that the authors reflect on that in their discussion in lines 220-227.
Author Response

(The authors gave the same response as above.)

Round 2
Reviewer 1 Report
I would like to express my appreciation for the thorough revisions made by the authors and the efforts in addressing the comments and concerns I raised during the initial review process. Thank you !